# Transport Characteristics of Fujifilm Ion-Exchange Membranes as Compared to Homogeneous Membranes АМХ and СМХ and to Heterogeneous Membranes MK-40 and MA-41

**DOI:** 10.3390/membranes9070084

**Published:** 2019-07-14

**Authors:** Veronika Sarapulova, Inna Shkorkina, Semyon Mareev, Natalia Pismenskaya, Natalia Kononenko, Christian Larchet, Lasaad Dammak, Victor Nikonenko

**Affiliations:** 1Department of Physical Chemistry, Kuban State University, 149 Stavropolskaya st., 350040 Krasnodar, Russia; 2Institut de Chimie et des Matériaux Paris-Est, UMR7182 CNRS—Université Paris-Est, 2 rue Henri Dunant, 94320 Thiais, France

**Keywords:** ion exchange membrane, electrospinning, electric conductivity, diffusion permeability, selectivity

## Abstract

Ion-exchange membranes (IEMs) find more and more applications; the success of an application depends on the properties of the membranes selected for its realization. For the first time, the results of a comprehensive characterization of the transport properties of IEMs from three manufactures (Astom, Japan; Shchekinoazot, Russia; and Fujifilm, The Netherlands) are reported. Our own and literature data are presented and analyzed using the microheterogeneous model. Homogeneous Neosepta AMX and CMX (Astom), heterogeneous MA-41 and MK-40 (Shchekinoazot), and AEM Type-I, AEM Type-II, AEM Type-X, as well as CEM Type-I, CEM Type-II, and CEM Type-X produced by the electrospinning method (Fujifim) were studied. The concentration dependencies of the conductivity, diffusion permeability, as well as the real and apparent ion transport numbers in these membranes were measured. The counterion transport number characterizing the membrane permselectivity increases in the following order: CEM Type-I ≅ MA-41 < AEM Type-I < MK-40 < CMX ≅ CEM Type-II ≅ CEM Type-X ≅ AEM Type-II < AMX < AEM Type-X. It is shown that the properties of the AEM Type-I and CEM Type-I membranes are close to those of the heterogeneous MA-41 and MK-40 membranes, while the properties of Fujifilm Type-II and Type-X membranes are close to those of the homogeneous AMX and CMX membranes. This difference is related to the fact that the Type-I membranes have a relatively high parameter *f*_2_, the volume fraction of the electroneutral solution filling the intergel spaces. This high value is apparently due to the open-ended pores, formed by the reinforcing fabric filaments of the Type-I membranes, which protrude above the surface of these membranes.

## 1. Introduction

In recent years, the fields of application of electrodialysis (ED), dialysis, and other processes with ion-exchange membranes (IEM) have expanded rapidly. In these fields, the environmentally friendly membrane technologies actively replace traditional chemical and physico-chemical methods of purification, concentration, conditioning, and separation of substances [1,2,3]. These applications include wastewater treatment [4], the extraction of antioxidants from wine-making waste or the increase in their concentration in fruit juices [5], as well as the recovery of dietary supplements and valuable medicinal products from wastes of agricultural products processing [6,7].

It is known that IEMs are the most expensive part of ED devices. Their contribution to the total cost of the process can reach 40–50% [8]. Therefore, the degree of implementation of technologies that use ion-exchange membranes is generally determined by IEM transport properties and cost.

Until recently, the structure of membranes mainly used in commercial applications could be reduced to two main types: (1) homogeneous at the nanoscale level (up to 100 nm) single-phase IEMs (Nafion, DuPont Co., Wilmington, CA, USA; MF-4SK, JSC NPO Plastpolymer, Saint Petersburg, Russia; and others) or two-phase IEMs (Neosepta AMX, CMX, Astom, Tokyo, Japan; CJMC-3, CJMA-7, Hefei ChemJoy Polymer Materials, Hefei, China; and others), which are traditionally called “homogeneous”; and (2) heterogeneous IEMs, which consist of micrometer (from 5 to 50 microns) ion-exchange polymer granules incorporated into an inert binder (MK-40, MA-40, Shchekinoazot, Russia; Ralex MH-PES, MEGA a.s., Czech Republic; FTAM-E, FTAM-A, FuMA-Tech GmbH, Ludwigsburg, Germany) [1,9,10].

To increase the mechanical strength of IEM, they are usually reinforced with nets or fabrics made of various polymers (capron, polyester, polyvinyl chloride, etc.). Several years ago, in the framework of the European project “Blue Energy”, Fujifilm (the Netherlands) set up an industrial production of fundamentally new IEMs [11,12]. The basis of these membranes is a three-dimensional structure of nanofibers made using the electrospinning method [13,14]. We denote these membranes by the term “electro-spun nanofiber ion exchange membranes”, EN-IEMs. These fibers, whose diameter ranges from 40 nm to 25 μm, simultaneously play the role of an inert filler and a reinforcing material. The space between the fibers is filled with a homogeneous ion-exchange aliphatic polymer. Note that the original ion-exchange membranes manufactured using the electrospinning method were used to produce electricity by the reverse electrodialysis method. An overview of these works can be found in [15,16,17]. The developers mainly sought to provide low electrical resistance while maintaining the high mechanical strength of the new membranes. Recently, a number of attempts have been made to apply EN-IEMs in fuel cells [18], to obtain ultrapure water [19], to remove fluoride and arsenic ions from natural waters [20], and others.

Especially attractive is their use in the processing of liquid media of the food and pharmaceutical industries, where the main priorities are the energy costs, the ability of membranes to transfer not only small mineral ions (Na^+^, Cl^–^), but also high-molecular compounds (polyphenols, amino acids, peptides, polysaccharides, and others), as well as their resistance to fouling by aromatic compounds [21,22,23]. However, the widespread introduction of relatively cheap membranes of this type is hampered by the lack of knowledge about transport characteristics of membranes manufactured by Fujifilm or experimental membranes, which have a similar structure [24,25]. To the best of our knowledge, the existing research is mainly devoted to the study of electric conductivity [26,27], the search for methods of the EN-IEM surface modification to control sedimentation [28], or to increase their specific selectivity to a specific type of counter-ions [29,30,31].

For example, in studies [26,27] it was shown that the electric conductivity of these membranes increases with an increase in their exchange capacity and decreases with an increase in the ionic strength of the external solution. The authors explain the last effect [27] by narrowing the conductive channels as a result of a decrease in the water content of the membranes.

The purpose of current work is a comprehensive study of the structural and transport characteristics (electric conductivity, diffusion permeability, transport numbers of co-ions and counterions) of three cation-exchange and three anion-exchange membranes manufactured by Fujifilm using the method of electrospinning. We will compare these data with the characteristics of known homogeneous and heterogeneous membranes and assess the prospects for using EN-IEMs in dialysis and electrodialysis processes.

## 2. Objects and Methods of Study

### 2.1. Membranes

Anion-exchange (АЕМ Type-I, AEM Type-II, AEM Type-Х) and cation-exchange (CEM Type-I, CEM Type-II, CEM Type-Х) membranes manufactured by Fujifilm (Tilburg, The Netherlands) were studied. The Fujifilm homogeneous membranes are based on a three-dimensional structure (substrate) of inert polyolefin fibers [12], which were obtained by electrospinning method [14]. The formation of fine fibers is carried out from a polymer solution in an organic solvent under the action of electric field of high strength. This solution is fed from the capillary needle and is accompanied by evaporation of the solvent. The thickness of the fibers is varied using different concentrations of polymer, the solution feed rate and the electric field strength [32,33,34]. The aerogel formed by the fibers is pressed to a predetermined thickness. The space between the fibers is filled with aliphatic polyamide, PA [23,26,35] functionalized by quaternary ammonium bases, –(CH_3_)_3_N^+^, in the case of anion-exchange membranes (AEMs) [36] or sulfo groups, –SO_3_^−^, in the case of cation-exchange membranes (CEMs) [36]. The chemical structure of cation-exchange and anion-exchange EN-IEM is shown in Figure 1.

Homogenous membranes AMX and СМХ (produced by Astom, Tokuyama Soda, Japan) are made by the “paste method” [37,38]. Initially, the paste contains styrene monomer with functional groups (which are subsequently grafted with ion-exchange groups), divinylbenzene (45–65%) as a crosslinking agent, a radical polymerization initiator and powdered polyvinyl chloride (45–55%). The paste is deposited on the reinforcing polyvinyl chloride fabric. The copolymerization is carried out before the sulfonation (CEM) or amination (AEM). According to [39] such membranes consist of two interpenetrating phases: ion exchange material and polyvinyl chloride, PVC, having a particle diameter of 100 nm or less. The structural inhomogeneities in the membrane volume does not exceed 1 micron [40]. An exception is the reinforcing fabric having fibers of 25–30 µm in diameter.

Heterogeneous membranes MA-41 and MK-40 (manufactured by JSC Shchekinoazot) are made by hot rolling of low-pressure polyethylene, PE (used as a binder), with AV-17 and KU-2 ion-exchange resin powders, respectively. The size of polyethylene and ion-exchange resin particles is in the range from 5 to 50 microns. Reinforcing nylon mesh (where the diameter of filaments is of 30–50 microns) is introduced by hot pressing [41].

Summation of basic materials and membrane manufacturing methods are presented in Table 1. Some of the characteristics of the swollen membranes are presented in Table 2.

### 2.2. Reagents

In experiments we use: distilled water (electric conductivity of 1.1 ± 0.1 μS cm^−1^; pH = 5.5; 25 °С), solid NaCl of the analytical grade (JSC Vecton), as well as natural dye—anthocyanin (Frutarom Etol d.o.o., Škofja Vas, Slovenia).

### 2.3. Methods of Membrane Characterization

Before conducting measurements, all membrane samples underwent standard salt preparation in NaCl solutions [9].

***The total exchange capacity*** (*Q_sw_*) is determined by the static acid-basic method [60], described in detail in Appendix A.

***Water content*** (*W*, %) of the membranes is determined as the mass ratio of water to swollen (sw) membrane by the gravimetric method, described in detail in Appendix A.

***The specific water-absorbing capacity*** (the number of moles of water per one mole of the fixed groups of the swollen membrane, *n_w_*) was determined by the formula:(1)$$n_{w}=\;\frac{W}{P_{H_{2}O}\cdot Q}$$
where $P_{H_{2}O}$ is the molar mass of water, equal to 18 g/mol; *Q* is the membrane exchange capacity, mol/g*_w_*.

***The thickness of the swollen membrane*** (*d*, μm) was controlled by a high-precision digital micrometer MKC-25 0.001 with an accuracy of 1 μm and with an error of 0.1 μm. The membrane thickness was obtained by averaging the results of 10 measurements made at various points of the sample under study.

***The water contact angles of the swollen membrane surface*** were determined by the method of a resting drop described in [61]. The distilled water served as a test liquid. The water contact angles were recorded in 20 s after the test drop touches the rough surface of the membranes.

***Visualization of the surface and volume of swollen IEM*** was carried out using an SOPTOP CX40M optical microscope (China) equipped with a digital eyepiece USB camera (5×, 10×, 20×, and 50× magnification). In order to increase the image contrast, Fujifilm membranes were placed in a solution containing a high molecular weight dye (anthocyanin) for 24 h before the measurements.

***The morphology and chemical composition*** of the air-dry IEM surface was studied using the AFM method in “contact” mode (JEOL 5400 microscope, Tokyo, Japan, JEOL 5400 software), as well as using a scanning electron microscope (MERLIN™, Carl Zeiss Microscopy GmbH, Jena, Germany) equipped with an energy dispersive spectrometer (accelerating voltage is 6 kV, working distance is 9.9 mm) and the device for X-ray fluorescence elemental spectroscopy (XRF). To improve the contrast of the obtained images and reduce the electric field strength, a thin layer of platinum nanoparticles was sprayed onto air-dry membrane samples. The method of surface metallization is generally accepted for SEM studies of ion-exchange materials [62]. It is known that nanoparticles of platinum and gold have the smallest size compared with other metals [63]. Therefore, the use of these nanoparticles practically does not distort the geometry of the surface under study. To estimate the fractions of secondary and tertiary amines in the composition of fixed groups of anion-exchange membranes, the ability of weakly basic amines to form chemically strong complex compounds with copper(II) ions was used [64]. To this end, before obtaining XRF, the samples were pretreated with a 2 M CuSO_4_ solution (pH = 4) for 48 h. Then samples were washed in distilled water to clean off the copper that did not enter the complex compounds. Studies of the structure and chemical composition were carried out on at least five different areas of each of the samples in order to ensure the reproducibility of the measurements. Dispersion in determining the elemental composition of membranes was ± 1 weight percent.

***The specific electric conductivity of IEM*** (*κ**) was determined by a differential method using a clip cell [65,66] and an immittance meter MOTECH MT4080 (Motech Industries Inc., Taiwan) at an alternating current frequency of 1 kHz. All samples were studied in 0.02–1.0 M NaCl solutions, starting from the lowest concentration.

The electric conductivity of membranes ($\kappa^{*}$) was determined by the equation:(2)$$\kappa^{*}=\frac{d_{m}}{R_{m+s}-R_{s}}$$
where *R_m+s_* is the resistance of membrane and solution; *R_s_* is the resistance of solution; and *d_m_* is the membrane thickness in the solution of given concentration.

***The diffusion characteristics of the IEM*** were investigated using a two-chamber flow cell [67]. The membrane separated two tracts: distilled water was pumped through one of them (tract I), and a NaCl solution of a given concentration was pumped through the other tract (II). Before the experiments, all samples were equilibrated with 0.02 M NaCl solution. The first measurements were carried out for the concentration of NaCl solution in tract II equal to 0.02 M. Then, this concentration was sequentially increased to 1.0 M. The membrane under investigation was in contact with each of the solutions for at least 5 h. The cell scheme, the methodology for conducting the experiment and processing the obtained data are described in detail in [68]. The error of determining the integral diffusion permeability coefficient, *P*, of membranes is equal to ±0.4 × 10^−8^ cm^2^s^−1^.

***The transport numbers*** of counterions ($t_{1}^{*}$) and co-ions ($t_{A}^{*}$) in the membranes under study were obtained using the concentration dependences of the specific conductivity (*κ**) and the integral diffusion permeability coefficient of the membranes (*P*). The method of their determination and formulas for the calculation are presented in [69]:(3)$$t_{1}^{*}=\frac{1}{2}+\sqrt{\frac{1}{4}-\frac{P^{*}F^{2}c}{2RT\kappa^{*}}}$$
(4)$$t_{A}^{*}=1-t_{1}^{*}$$
where *F* is the Faraday constant; *R* is the gas constant, *T* is the temperature; *C* is the electrolyte concentration (NaCl), *P** is the local diffusion permeability coefficient of the membrane, which is connected to integral diffusion permeability coefficient by the relation:(5)$$P^{*}=P+C\frac{dP}{dC}$$

The value of the local diffusion permeability coefficient was determined taking into account Equation (1) as [70]:(6)$$P^{*}=P\left(\beta +1\right)$$

The coefficient *β = dlgP/dlgc* was found as the slope of the concentration dependence of the integral diffusion permeability coefficient, presented in logarithmic coordinates. In accordance with the generally accepted definition [71], the values of the counterion transport numbers through the IEM give an idea of the membranes’ selectivity.

***The standard contact porosimetry*** was used to estimate the integral volume of sorbed water (*V*) and the number of water molecules per AEM fixed group as functions of the effective pore radius (*r*). This method is described in detail in the works [72,73]. The essence of the method consists in measuring the equilibrium curve of the integral volume of sorbed water (or other liquid) of the sample under investigation, which is clamped between two reference porous samples with a known pore distribution. Porometric curves for reference samples are obtained in independent experiments, for example, using the method of mercury porosimetry. A controlled change in the water content of the samples is carried out by evaporation.

The effective pore radii that dominate in the membrane under study were determined from the values recorded at the inflection points of the porogram presented in the coordinates *V–r*. The total water content in the pores of this and a smaller diameter was estimated by the values of *V*, in the areas of the plateau of the porograms that follow the inflection points. The total water content in the test sample (*V*_0_) was found by the difference in weight of the completely swollen and absolutely dry membranes.

## 3. Results and Discussions

### 3.1. Structural Characteristics of the Investigated Membranes

Figure 2 and Figure 3 show the SEM and AFM images of the EN-IEM air-dry samples under study. The points in Figure 2 indicate where the elemental composition of the surface was determined using XRF. Optical images of swollen AMX Type I and MA-41 membrane samples are presented in Figure 4 and Figure 5. Optical images of dry AMX Type I, AMX Type II, CMX Type I, and CMX Type II membrane samples are presented in Appendix A. The randomly arranged reinforcing fibers of thickness from 10 to 25 microns occupy a significant part of the EN-IEM volume. Some of them protrude slightly above the surface. Cracks of elongated shape, apparently, are formed during drying of the samples before the SEM.

In the SEM (Figure 2) and AFM (Figure 3) images of the EN-IEM surfaces, round holes are observed. The diameter of these holes varies from 3 to 25 microns (Figure 3). AFM images show that the average depth (R_z_) of these holes is about 100 nm. These holes are visible on the surface of all investigated membranes (Figure 2).

The largest number of holes per surface unit is observed on the AEM Type I, CEM Type I, and CEM Type II membranes. Similar holes were found on membranes of this type by Y. Zhu et. al. [35], who suggested that these holes are formed at the stage of membrane manufacturing, and their presence causes the deterioration of the transport characteristics of EN-IEMs. However, it is seen in the AFM (Figure 3) and SEM (Figure 2) images that most of these defects have a small depth (about 50–200 nm) and only some of them have a depth greater than 20 μm. Therefore, a noticeable effect of these holes on the transport properties of EN-IEMs seems unlikely. Optical microscopy data (Figure 4) allows one to conclude that extended (lengthy) macropores in some samples of EN-IEM (Type-I, Type-II) do exist, but they are formed at the interface of the polymer fiber with the ion-exchange material. In the case of swollen samples in equilibrium with sodium chloride solution, these pores are filled with an optically transparent equilibrium solution (Figure 4a).

After the contact of EN-IEM samples with a solution that contains a high molecular weight dye, its molecules do not penetrate into the microporous ion-exchange material, but transfer through large pores and are visualized as dark bands recorded at the interface between the fibers and ion-exchange material, as in the case of AEM Type-I shown in Figure 4b. Similar dark bands appear around the fibers in the swollen AEM Type-II membrane (Appendix A).

The occurrence of diffusion pathways along the filaments of reinforcing net of swollen heterogeneous membranes can be observed although in the process of moisture evaporation from these membranes, as in the case of MA-41 membrane, Figure 5. Water has a higher optical density than air. Therefore, the near-surface membrane areas, where water is partly evaporated, should have a darker color. On the surface of a sample, which was in contact with air for a short time (Figure 5a), there are no dark areas. However, they appear after a few hundred seconds, especially at the intersections of the filaments of the reinforcing net, which are closest to the surface (Figure 5b). Then the dark areas spread along the entire length of the reinforcing net filaments and are localized at the borders between the filaments and other membrane components, ion-exchange resin particles, and polyethylene binder (Figure 5c), as well as at the boundaries between the resin particles and polyethylene binder in places where these particles reach the surface (Figure 5c). It is important to note that in the case of drying of homogeneous AMX and CMX membranes, such pores are not visualized.

The results obtained using standard contact porosimetry method show that the nanoporous structure of the MK-40 and MA-41 membranes is similar to that of the homogeneous membranes produced by the paste method (CMX, AMX). However, the MK-40 and MA-41 membranes have macropores with a radius of more than 35 nm, while the homogeneous membranes do not have such pores (Figure 6).

Indeed, in the range of effective pore radii from 1 to 15 nm, the porometric curves for heterogeneous (MA-41) and homogeneous (AMX) membranes are almost identical. The dominant effective pore radii, which correspond to the inflection points in the curves, are 3 and 13 nm. Such pore sizes are typical for homogeneous ion-exchange materials [73]. A similarity in porometric curves is explained by the fact that both membranes are made of styrene copolymer with divinylbenzene and have the same fixed groups. The dry MA-41 membrane is characterized by a higher concentration of fixed groups (exchange capacity), so water content in its micropores (*r* < 3 nm) and mesopores (*r* < 13 nm) is higher as compared to AMX. Both membranes have pores with a radius of about 35 nm, which, apparently, can be attributed to defects in the structure of ion-exchange polymers, but their contribution to the total water content is insignificant, especially in the case of AMX. Further, the AMX porometric curve comes to a plateau, and the MA-41 porometric curve still has at least two inflections which indicate the presence of larger pores. The first of them (the effective radius is about 100–150 nm) could be attributed to macropores formed at the places of contact of resin particles with an inert reinforcing binder (polyethylene). Such pores were found, for example, in the images of the surface of swollen MK-40 and MA-41 membranes, using the low-vacuum SEM method [55].

The larger pores (>3000 nm) can be visualized by the optical method as dark areas (Figure 5). They are formed between the nylon reinforcing net and the resin particles as well as the polyethylene binder (Figure 5). According to our estimates, made by the analysis of porometric curves (Figure 6), both types of macropores (>100 nm) contain up to 25% of the water in MK-40 and MA-41 membranes. These voids appear when the membrane is dried. In the swollen state, the volume fraction of macropores may be evaluated as about 10% [74]. The absence of such large pores appearing when drying in homogeneous AMX (Figure 6) and CMX membranes is apparently associated with stronger adhesion of the reinforcing net and inert binder, which are produced from the same material, polyvinyl chloride.

The elemental composition of the investigated EN-IEMs is obtained using XRF and presented in Table 3. The reinforcing fibers and ion-exchange material are studied separately. In the case of fibers (the lines in Table 3 are indicated by odd numbers), the composition corresponds to polyolefin; in the case of ion-exchange material (indicated by even numbers), to polyamide (Figure 1), from which the membranes are made [36]. The sulfonic fixed groups of the CEM contain sulfur. The studied samples were in the H^+^ form. These counterions are not identified by XRF. The presence of fixed amino groups in AEM leads to an increase in nitrogen concentration as compared to CEM.

Chlorine, which is found on the surface of AEM, is due to Cl^−^, a counterion. The percentage of nitrogen and chlorine (in AEMs), as well as sulfur (in CEMs), increases in proportion to the growth in the exchange capacity of dry membranes (Table 3). It should be noted that all manufacturers and most of the researchers indicate that the functional groups of the studied AEMs are quaternary amines. However, these membranes contain secondary and tertiary amines [75]. According to XRF data (Appendix A), copper (ions Cu^2+^), which is an indicator of the presence of weakly basic groups, is evenly distributed over the surface and volume of ion exchange materials and its mass fraction in the case of unused AMX, MA-41, AEM Type-II and AEM Type-X does not exceed 1% (see Appendix A for an AEM Type-X membrane). An exception is the AEM Type-I membrane, the mass fraction of copper which reaches 3% (Appendix A). It is known [76] that the thermal destruction and/or nucleophilic attack of quaternary amines with hydroxyl ions (which are always present in aqueous solutions) is the cause of the secondary and tertiary amines formation. The transformation of strongly basic amines into weakly basic ones can occur both at the manufacturing stage and during the storage of membranes in aqueous solutions. This is probably why the information on the content of these groups in anion-exchange membranes, for example, in AMX, differs significantly [31,75]. Given that all membranes, with the exception of AMX, are stored in a dry state, it can be assumed that the higher content in AEM Type-I of secondary and tertiary amines is due to the special conditions of its manufacture.

Thus, both the studied heterogeneous membranes and EN-IEMs contain extended macropores, which are formed at the points of contact of the fibers (Type-I and Type-II) and the reinforcing net (MA-41, MK-40) with the ion-exchange material. There are no such pores in the AMX and CMX membranes.

In general, the chemical composition of the investigated membranes corresponds to that declared by the manufacturers. All CEM contain sulfonic functional groups. All AEM contain strongly basic quaternary amines. The AEM Type-I membrane is characterized by the greatest portion of weakly basic functional groups in comparison with other studied AEM.

### 3.2. Membrane Transport Characteristics

***The specific conductivity of the membranes***. Figure 7a and Figure 8a show the concentration dependences of the specific conductivity of the investigated membranes in sodium chloride solutions. In Figure 7b and Figure 8b the dependences of the specific conductivity of these membranes on the specific conductivity of NaCl solutions with which these membranes are equilibrated are shown in logarithmic coordinates. Such data processing allows determining the volume fractions of the gel phase (*f*_1_) and the phase of the electroneutral solution filling the intergel spaces (*f*_2_) of the investigated membranes according to the microheterogeneous model [59]. This model, in the simplest case, considers an IEM as a two-phase system. The gel phase is a microporous swollen medium, which includes a polymer matrix that carries fixed charged groups, as well as a charged solution of mobile counterions (and the co-ions in a smaller amount), which compensates the charge of fixed groups. The reinforcing fibers (fabric), inert filler or binder are considered as parts of the polymer matrix. The electroneutral solution filling the intergel spaces, which are the central part of the mesopores and the structural defects of the membrane, is assumed identical to the external solution.

The simplified equation connecting the conductivity of the membrane (*κ**) with the conductivities of the gel phase ($\bar{\kappa }$) and the intergel solution (*κ*), in the framework of the microheterogeneous model, reads:(7)$$\kappa^{*}={\bar{\kappa }}^{f_{1}}\kappa^{f_{2}}$$
According to Equation (7), the lg*κ** vs. lg*κ* dependence should be a straight line, the conductivity of the gel phase being considered as constant. The slope of this line is equal to $f_{2}$ ($f_{1}$ + $f_{2}$ = 1), the other coefficient of this linear dependence allows finding the value of $\bar{\kappa }$.

Knowing the value of $\bar{\kappa }$, it is possible to estimate the counterion diffusion coefficient, $\bar{D_{1}}$, in the gel phase (when neglecting the contribution of co-ions to the conductivity):(8)$$\bar{D_{1}}=\left(RT/F^{2}\right)\left(\bar{\kappa }/\bar{Q}\right)=\left(RT/F^{2}\right)\left(\bar{\kappa }f_{1}/Q\right)$$

Here *R* is the gas constant; *T* is the temperature; *Q* is the membrane exchange capacity (mmol cm^−3^ membrane), $\bar{Q}=Q/f_{1}$ is the exchange capacity of the gel phase of the swollen membrane (mmol cm^−3^ gel).

Note that the concentration at which the conductivities of the membrane, gel phase and external solution are equal is called the isoconductivity point, the corresponding concentration is denoted as *C*_iso_. At concentrations less than *C*_iso_ (for studied IEM, *C*_iso_ is in the range of 0.03–0.10 M solutions), $\bar{\kappa }$ is greater than *κ*; at *C* > *C*_iso_, $\bar{\kappa }$ is lower than *κ*.

The values of *f*_2_, $\bar{\kappa }$, $\bar{Q}$, and $\bar{D_{1}}$ found as described above are presented in Table 4. In the case of studied membranes with an aromatic ion exchange matrix, these values are in good agreement with the literature data (Table 2 and Table 4). Some differences, apparently, are due to the fact that the properties of all membranes vary from one surface site to another; they depend on the peculiarities of membrane preparation as well as the storage conditions.

The values of $\bar{\kappa }$ increase in the series CEM Type-II < MK-40 < CMX ≈ CEM Type-I < CEM Type-X (for CEMs) and AMX ≈ AEM Type-II < MA-41 < AEM Type-X < AEM Type-I (for AEMs) in the same order as the exchange capacity of the gel phase (Table 4).

An exception is the AEM Type I membrane. The values of $\bar{\kappa }$ and $\bar{D_{1}}$ for this membrane are greater as compared to AEM Type-X at close values of $\bar{Q}$ for both membranes. This experimental fact is probably explained by a higher content of secondary and tertiary amino groups in the AEM Type-I membrane (Appendix A). Our studies show that the treatment of a weakly basic heterogeneous MA-40 membrane (JSC Shchekinoazot, Russia) with a modifier, which leads to shielding of secondary and tertiary fixed amino groups with quaternary amines, causes a decrease in the conductivity of this membrane in 1 mol dm^−^^3^ NaCl solution from 6.9 to 4.5 mS cm^−^^1^, while *f*_2_ of the pristine and modified membranes do not change significantly [78].

As one would expect, in the moderately concentrated solutions (0.1 M < *C* < 1.0 M), the rate of increase in conductivity of all investigated membranes with respect to the conductivity at the *C = C*_iso_ is determined by *f*_2_: the higher this value, the stronger the conductivity increases with increasing concentration of the external solution.

***Diffusion permeability of ion-exchange membranes***. Figure 9 shows the concentration dependences of the integral diffusion permeability coefficient, *P*, of investigated membranes in NaCl solutions. It is known [71] that the value of *P* (or the value of differential diffusion permeability *P**, Equation (5)) is mainly determined by the rate of co-ion diffusion, which depends on the gel phase exchange capacity ($\bar{Q}$), the co-ion concentration (${\bar{C}}_{A}$) and diffusion coefficient (${\bar{D}}_{A}$) in this phase, the volume fraction of electroneutral solution (*f*_2_) and the relative disposition of the gel and intergel solution phases, parameter α [59]:(9)$$P^{*}=2t_{1}^{*}{\left[f_{1}{\left({\bar{D}}_{A}^{}{\bar{C}}_{A}\right)}^{\alpha }+f_{2}{\left(D_{A}^{}C\right)}^{\alpha }\right]}^{1/\alpha }RT/C$$

The value of $\bar{Q}$ determines the degree of electrostatic (Donnan) exclusion of co-ions from the gel phase and determine the value of the counterion transport number ($t_{1}^{*}$); *f*_2_ determines the intergel solution regions where this electrostatic exclusion is absent. The greater $\bar{Q}$ ($t_{1}^{*}$) and the lesser *f*_2_, the lesser the diffusion permeability of the membrane. The value of *P* increases when the gel and intergel solution phases are disposed so that the intergel solution forms a continuous pathway for co-ions, that is, the phases are mainly in parallel to each other (α approaches 1) and the contribution of the series disposition of phases is low. Dilution of the solution should lead to a decrease in the diffusion permeability of the membranes due to increase in electrostatic (Donnan) exclusion of co-ions from the gel phase [71].

The contribution of macropores to the membrane transport properties, in particular, to diffusion permeability, depends on the amount of these pores per unit volume and their disposition in space. These pores are mainly formed at the interfaces of different structure elements of the membrane, namely, ion-exchange material, binder and reinforcing fabric. The highest volume fraction of macropores expressed by the highest value of *f*_2_ is in the heterogeneous MK-40 and MA-41 membranes (about 0.2); the Type-I cation- and anion-exchange Fujifilm membranes have *f*_2_ close to 0.15; the other studied membranes, Fujifilm and Neosepta, have this parameter between 0.06 and 0.11 (Table 4). The value of *P* is the lowest for the membranes with small *f*_2_, it is essentially higher for the MK-40, MA-41 and Type I cation- and anion-exchange Fujifilm membranes. However, it can be remarked that the values of *P* are close for the last four membranes, while *f*_2_ is noticeably different. This can be explained by higher fraction of parallel disposition of macropores (higher α) in the Type I membranes in comparison with the MK-40 and MA-41 membranes. The reason for higher α could be in the appearance of long open-ended macropores along the fiber of the reinforcing, which protrude over the membrane surface. These pores form continuous pathways not interrupted by the elements of the gel phase, which is almost impermeable for co-ions. SEM images show that in the case of Type-I membranes (Figure 2a,b) there is a great number of protrusions of the reinforcing fibers over their surface. This hypothesis is supported by the literature data [35], claiming that the selectivity of the CEM Type-I membrane increases when the membrane surface on the receiving side is covered with a film of ion-selective material. The number of these protrusions is much lower in the case of heterogeneous MK-40 and MA-41 membrane (Figure 2h), where the filaments of the fabric are immersed into the membrane body. Similarly, the fibers of the reinforcing fabric are immersed within the Neosepta membranes (Figure 2g). In addition, there is a good adhesion between the reinforcing fabric and ion-exchange material of the Neosepta membranes due to the similarity of their chemical composition (see Section 2.1). As for the Type-II and Type-X of EN-IEMs, as Figure 2c–f show, the number of fibers and their protrusions on the surface is essentially lower than in the case of the Type I membranes.

*Counterion and co-ion transport numbers.*Figure 10 shows the concentration dependences of the (real) transport numbers ($t_{1}^{*}$) of counterions in the investigated IEMs, which are found from the values of their conductivity and diffusion permeability using the approach [69] described above in Section 2.3, Equation (3). Table 5 summarizes $t_{1}^{*}$, determined by the authors of this paper and by other researchers, as well as the apparent transport numbers ($t_{1app}^{*}$) from literature, determined using the potentiometric method [9,79]. As a rule, $t_{1app}^{*}$ values are indicated in the catalogs of manufacturers.

The relationship between the real and apparent transport numbers is determined by the Scatchard equation [9,80]:(10)$$t_{1app}^{*}=t_{1}^{*}-z_{1}n_{1}m_{s}M_{w}t_{w}$$
where *m_s_* is the molality of the solution, *M_w_* is the molar mass of water, *z*_1_ and *n*_1_ are the charge and stoichiometric numbers of counterion, $t_{w}$ is the water transport number showing how many moles of water are transported when 1 Farad of electric charge passes through the membrane.

The real (electromigration) ion transport number is a fundamental characteristic of the selective properties of ion-exchange membranes. The value of $t_{1app}^{*}\;$ depends not only on the real transport number $t_{1}^{*}$, but also on the membrane electroosmotic permeability, which determines the value of $t_{w}$, Equation (10). The values of $t_{w}$, found using Equation (10) and known values of $t_{1}^{*}$ and $t_{1app}^{*}$ are presented in Table 5.

The determined values of the real counterion transport numbers in homogeneous (AMX, CMX) and heterogeneous (MA-41, MK-40) aromatic membranes are in good agreement with the results obtained by other researchers [69]. In the case of relatively concentrated solutions (*C* = 1.0 M), the permselectivity of the CEM Type-II and CEM Type-X cation-exchange membranes (Figure 10a) and the AEM Type-II and AEM Type-X anion-exchange membranes is close to that determined for the homogeneous CMX and AMX membranes (Astom, Japan) with an aromatic matrix. The permselectivity of the Type-I cation- and anion-exchange membranes is comparable to that of heterogeneous membranes MK-40, MA-41 (Shchekinoazot, Russia). In all cases, the co-ion transport numbers in the 1.0 M NaCl solution do not exceed the value of 0.03.

## 4. Conclusions

A comparative analysis of the concentration dependeces of the conductivity, diffusion permeability, real and apparent ion transport numbers in cation- and anion-exchange membranes produced by three manufacturers was conducted. The microheterogeneous model [59] was applied to the interpretation of the experimental results.

The studied membranes involves (1) Neosepta AMX and CMX membranes (Astom, Japan), which are traditionally called homogeneous; (2) heterogeneous IEMs, which consist of micrometer (from 5 to 50 microns) ion-exchange resin particles, incorporated in an inert binder (MK-40, MA-40, JSC Shchekinoazot, Russia), and (3) Type-I, Type- II, and Type-X anion-exchange and cation-exchange membranes by Fujifilm, the Netherlands, which are based on a three-dimensional structure of interlaced fibers with a diameter from 40 nm to 25 μm; the space between the fibers is filled with a homogeneous ion-exchange polymer. Membranes (1) and (2) contain ion exchange materials, which are based on copolymers of polystyrene with divinylbenzene. They are reinforced with fabric (1) or net (2), made of polymeric materials. The basis of ion exchange membrane materials (3) is hydrophilic aliphatic polyamide. The three-dimensional structure of interwoven fibers simultaneously performs the functions of both a filler and a reinforcing material.

In NaCl solutions, all types of cation- and anion-exchange membranes manufactured by Fujifilm have a high permselectivity sufficient for use in electrodialysis of dilute solutions. The real counterion transport numbers in the Type-II and Type-X membranes are close to the values characteristic to CMX and AMX membranes, which are widely used in various electromembrane processes. In the Type-I membranes, transport numbers are similar to those determined in heterogeneous membranes MK-40 and MA-41.

The concentration dependences of the conductivity and diffusion permeability of Fujifilm membranes generally follow the same laws that are known for well-studied homogeneous (CMX, AMX) and heterogeneous (MK-40, MA-41) membranes. The structure-property relationships of all these membranes are well described and predicted by the microheterogeneous model.

One of the features in the behavior of Fujifilm membranes is in the fact the Type-I membranes show a high diffusion permeability and low permselectivity comparable to those of the heterogeneous MK-40 and MA-41 membranes. The reason, apparently, lies in the fact that the manufacturing of these membranes allows the formation of open-ended macropores on the membrane surface due to the protrusions of the fabric filaments above the surface. In the CMX and AMX membranes, large extended pores are practically absent, because the reinforcing fabric and filler of the ion exchange material are made of the same polymer (polyvinyl chloride) and have a good adhesion to each other.

In the case of MK-40, MA-41, the effect of these pores on the transport characteristics is essentially lower than in the case of the Type-I membranes, since the reinforcing net is immersed inside the membrane bulk.

## Figures and Tables

**Figure 1 membranes-09-00084-f001:**
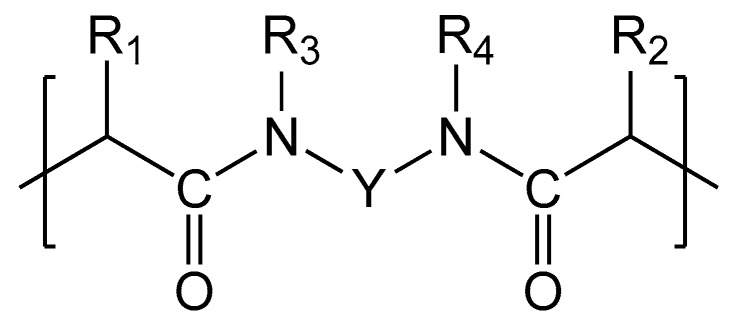
Chemical structure of cation- and anion-exchange EN-IEMs: R_1_ and R_2_ are each independently H or methyl; R_3_ and R_4_ are each independently H or alkyl, or R_3_ and R_4_ together with the N atoms to which they are attached, and Y form an optionally substituted 6- or 7-membered ring; Y is an optionally substituted and optionally interrupted alkylene or arylene group; provided that the structural unit has 1, 2, 3, or 4 sulphonic acid (CEM) or quaternary ammonium (AEM) groups.

**Figure 2 membranes-09-00084-f002:**
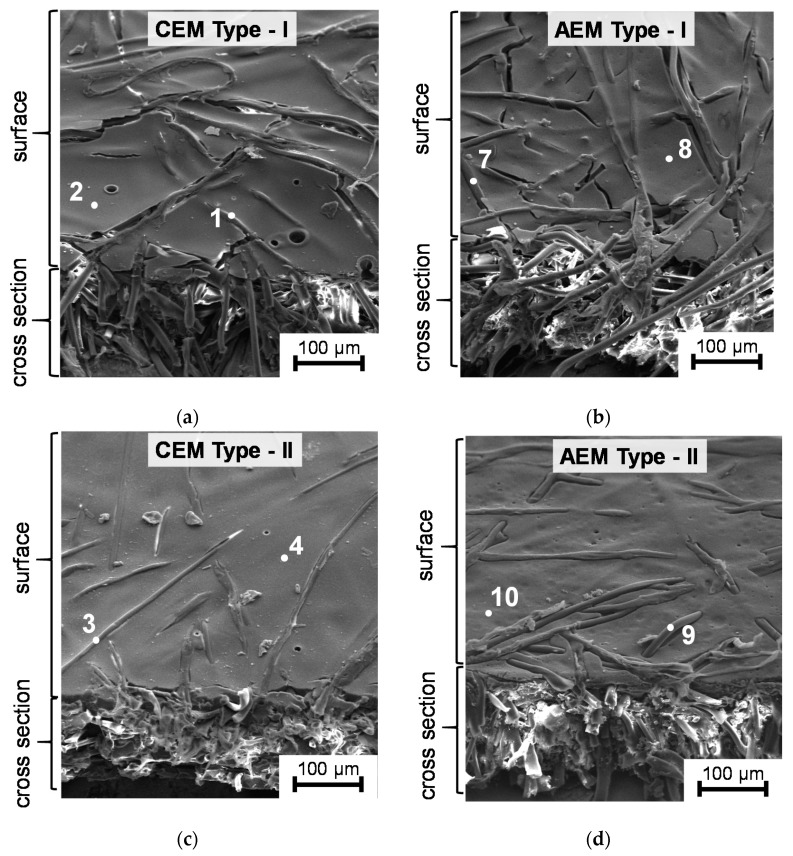
SEM images of the surface and cross sections of cation-exchange (**a**,**c**,**e**) and anion-exchange (**b**,**d**,**f**) IEM, manufactured using the electrospinning method, as well as Neosepta CMX (**g**) and heterogeneous MK-40 (**h**) membranes. The places where the element analysis was performed using the XRF are marked with white points.

**Figure 3 membranes-09-00084-f003:**
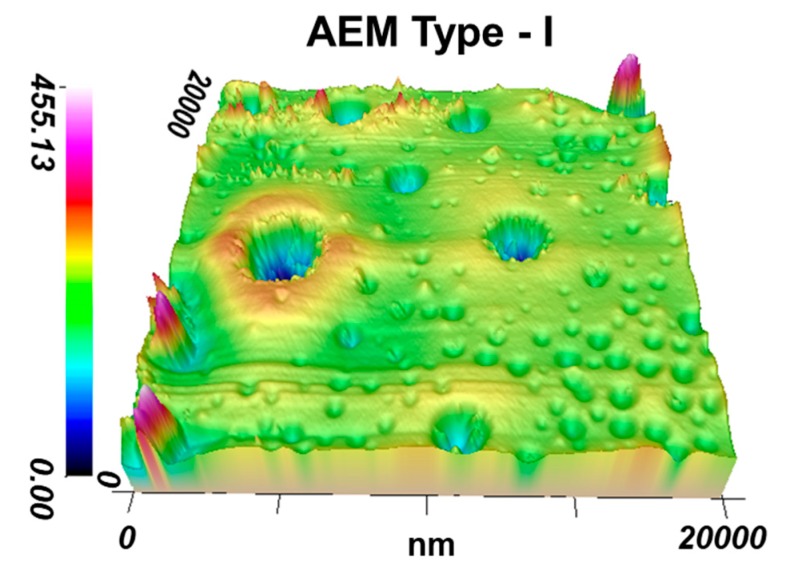
AFM image of air-dried sample of the AEM Type-I membrane.

**Figure 4 membranes-09-00084-f004:**
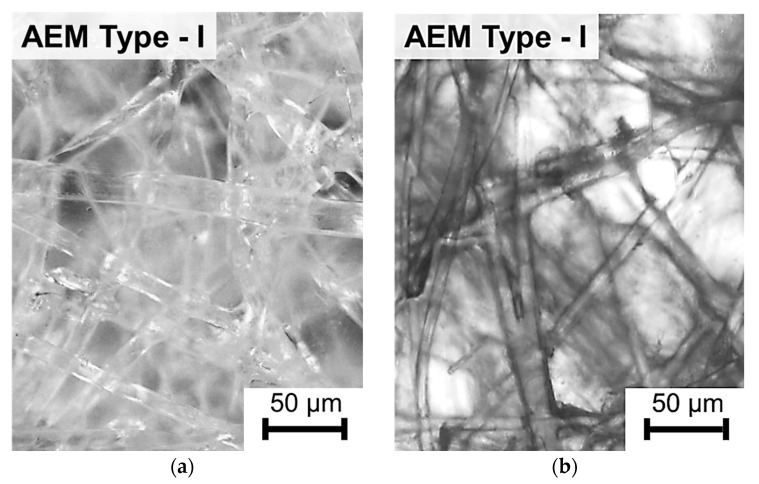
Optical images of the swollen AEM Type-I membrane. Before the study, samples were soaked in 0.02 М NaCl solution for 24 h (**a**) and in the 0.02 М NaCl solution where the high molecular weight dye (anthocyanin) was added (**b**).

**Figure 5 membranes-09-00084-f005:**
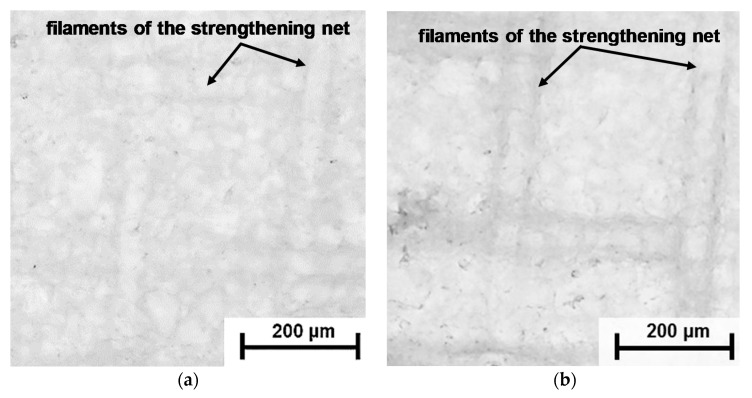
Optical images of the МА-41 membrane surface obtained using transmitted light mode in moments of time 10 s (**a**), 350 s (**b**), 720 s (**c**), and 900 s (**d**) after the beginning of contact of the swollen sample with the air.

**Figure 6 membranes-09-00084-f006:**
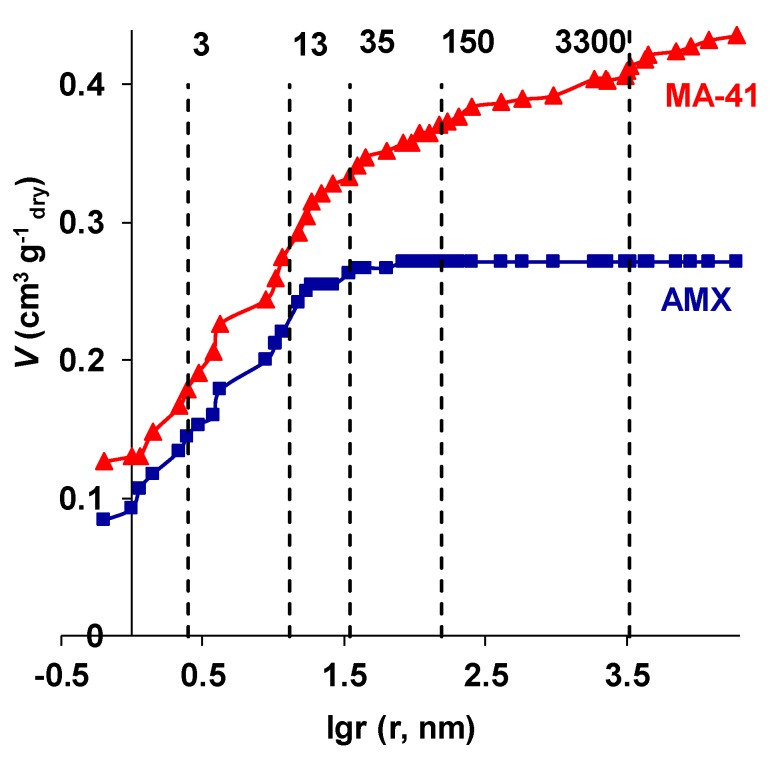
The dependence of the volume of sorbed water (*V*) on the effective pore radius (r) in an AMX and a МА-41 membrane. The vertical dashed lines indicate the inflection points on the curves; these points correspond to the radii of the pores whose fraction in the membranes is relatively high. The values (in nanometers) of these radii are shown by the numbers near the dashed lines.

**Figure 7 membranes-09-00084-f007:**
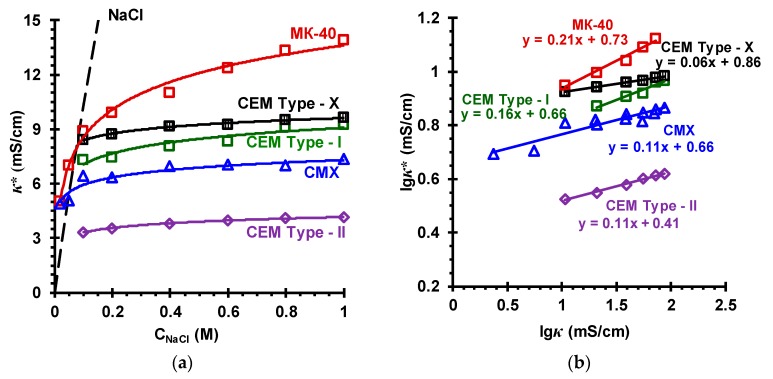
Concentration dependences of the conductivity of different cation-exchange membranes, *κ**, in NaCl solutions (**a**) and the lg*κ** vs. lg*κ* dependences of these membranes in NaCl solutions (**b**). The names of the membranes are indicated near the curves, which fit the experimental data. The dashed line in the figure (**a**) is the concentration dependence of the NaCl solution conductivity.

**Figure 8 membranes-09-00084-f008:**
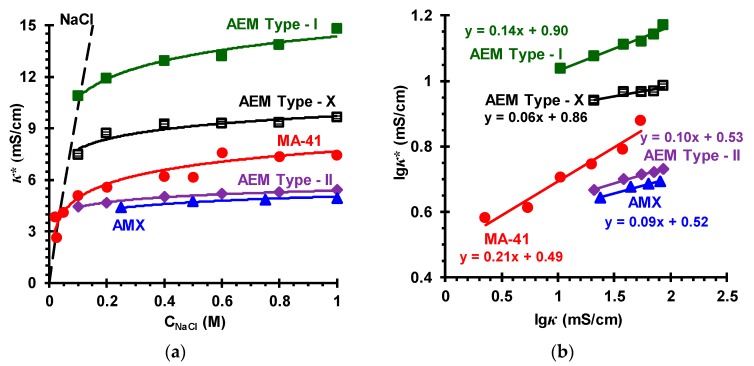
Concentration dependences of the conductivity of different anion-exchange membranes, *κ**, in NaCl solutions (**a**) and the lg*κ** vs. lg*κ* dependences of these membranes in NaCl solutions (**b**). The names of the membranes are indicated near the curves, which fit the experimental data. The dashed line in the figure (**a**) is the concentration dependence of the NaCl solution conductivity.

**Figure 9 membranes-09-00084-f009:**
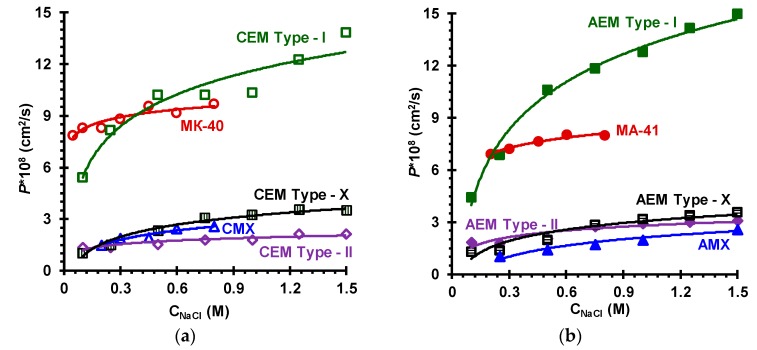
Concentration dependences of the integral diffusion permeability coefficient for cation-exchange (**a**) and anion-exchange membranes (**b**) in NaCl solutions.

**Figure 10 membranes-09-00084-f010:**
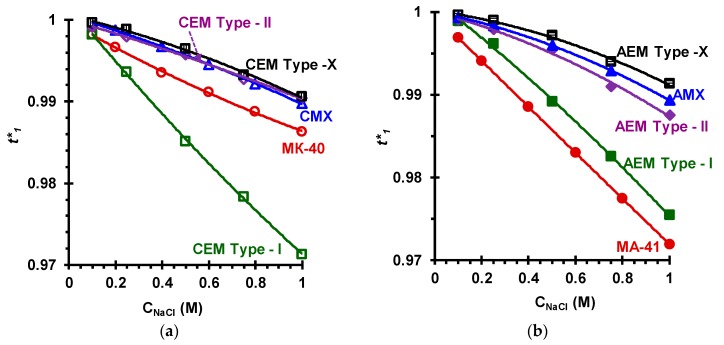
The concentration dependence of the real counterion transport numbers in the cation- (**a**) and anion-exchange (**b**) membranes.

**Table 1 membranes-09-00084-t001:** Summation of basic materials and membrane manufacturing methods.

Membranes	Type	Ion Exchange Matrix	Fixed Groups	Inert Binder	Reinforcing Material	Manufac-Turing Method
MA-41	heterogeneous	DVB + PS	*	PE	nylon mesh	hot rolling
MK-40			**			
AMX	homogeneous		*	PVC	PVC fabric	“paste” method
CMX			**			
АЕМ Type-I, AEM Type-II, AEM Type-Х		PA	*	-	3D polyolefin fibers structure	filling the voids of the 3D fiber structure with ion-exchange material
CEM Type-I, CEM Type-II, CEM Type-Х			**			

* Mainly quaternary ammonium bases, –(CH_3_)_3_N^+^, and a small amount of secondary and tertiary amines. ** Sulfo group –SO_3_^−^. DVB + PS is a copolymer of polystyrene and divinylbenzene; PE is low-pressure polyethylene; PA is polyamide; PVC is polyvinyl chloride.

**Table 2 membranes-09-00084-t002:** Some of the characteristics of the membranes under study. Our data are given without reference.

Membranes	Thickness of Air-Dried Membrane, Microns	* Thickness of Wet Membrane, Microns	Exchange Capacity, mmol g^−1^ wet	Density, g cm^−3^ wet	Water Content, g Н_2_О/g wet, %	Water Content, mol H_2_O/mol Functional Groups
Cation-Exchange						
CEM Type-I	120 ± 5 115 [42]	140 ± 10	1.43 ± 0.05	1.15	29 ± 5	11.3 ± 1
CEM Type-II	165 ± 5	180 ± 10 179 [43]	1.35 ± 0.05	1.13	25 ± 2	10.3 ± 1
CEM Type-X	125 ± 5 125 [42]	130 ± 5	1.67 ± 0.05	1.20	21 ± 5	7 ± 1
CMX	155 ± 5 155 [42]	170 ± 5 175 ± 6 [44] 163 [45]	1.61 ± 0.05 1.65 ± 0.05 [46] 1.62 ± 0.04 [31,47]	1.32 1.19 [48] 1.26 [49]	22 ± 3 19 [50] 23 [51]	8 ± 1 (7.5) 7.62 [52,53] 9 [54]
MK-40	440 ± 10	520 ± 20 520 ± 10 [55]	1.43 ± 0.08 1.52 ± 0.08 [55] 1.69 [48]	1.18 1.13 [48] 1.19 [55]	26 ± 5 34.7 [56] 33 ± 1 [55]	10 ± 1 12.1 ± 1 11.8 [57]
Anion-Exchange						
AEM Type-I	120 ± 5 115 [42]	125 ± 5	1.50 ± 0.05	1.06	8 ± 2	3.3 ± 1
AEM Type-II	165 ± 5 130 [19]	175 ± 5 175 [43]	1.08 ± 0.05	1.06	10 ± 5	5.5 ± 1
AEM Type-X	115 ± 5 125 [42]	120 ± 5	1.50 ± 0.05	1.08	23 ± 2	8.7 ± 1
AMX	125 ± 5	135 ± 5 141 ± 6 [23]	1.23 ± 0.05 1.25 [47]	1.22 1.10 [58]	14 ± 2 16 [50]	6.5 ± 1 6.1 [23] 7.8 [54]
MA-41	430 ± 10	450 ± 50 530 ± 20 [55]	1.22 ± 0.06 1.18 ± 0.06 [55] 1.25 [46]	1.06 1.14 [59]	30 ± 2 30 [56] 35 ± 2 [55]	8.7 ± 1 13 [56] 7.1 [23]

* Membrane equilibrated with 0.02 M NaCl solution.

**Table 3 membranes-09-00084-t003:** The elemental composition (in weight percent) of EN-IEM and their exchange capacity in the air-dry state.

Membranes		Elemental Composition (%)					*Q*, mmol g^−1^_dry_
		Carbon (С1s)	Oxygen (О1s)	Nitrogen (N1s)	Chlorine (Cl2p)	Sulfur (S2p)	
Cation-Exchange Membranes							
CEM Type-I	* 1	99	1	-	-	-	2.02 ± 0.05
	2	54	26	11	-	9	
CEM Type-II	3	99	1	-	-	-	1.81 ± 0.05
	4	50	32	10	-	8	
CEM Type-X	5	100	-	-	-	-	2.12 ± 0.05
	6	52	27	8	-	13	
Anion-Exchange Membranes							
AEM Type-I	7	99	1	-	-	-	1.65 ± 0.05
	8	57	21	16	6	-	
AEM Type-II	9	99	1	-	-	-	1.21 ± 0.05
	10	63	19	14	4	-	
AEM Type-X	11	100	-	-	-	-	1.96 ± 0.05
	12	75	5	13	7	-	

* The number in the table corresponds to the number of the membrane area indicated in Figure 2 which was analyzed using XRF. Even numbers denote the ion exchange material, odd numbers denote the reinforcing fibers.

**Table 4 membranes-09-00084-t004:** The values of conductivity at the isoconductivity points, volume fraction, exchange capacity of and counterion diffusion coefficient in the gel phase of the investigated membranes in NaCl solution.

Membranes	*f* _2_	$\bar{\kappa }$, mS cm^−1^	$\bar{Q}$, mmol cm^−3^_wet gel_	$\bar{D_{1}}10^{10}$, m^2^ s^−1^
Cation-Exchange Membranes				
CEM Type-I	0.16 ± 0.1	6.1 ± 0.1	1.65 ± 0.05	0.67
CEM Type-II	0.11 ± 0.01	2.8 ± 0.1	1.53 ± 0.05	0.36
CEM Type-X	0.06 ± 0.01	8.2 ± 0.1	2.01 ± 0.05	0.97
CMX	0.11 ± 0.03 0.06 [57] 0.10 [46]	5.5 ± 0.1 6.72 [57] 8.79 [46]	2.13 ± 0.05	0.63
MK-40	0.21 ± 0.03 0.25 [46] 0.18 [57] 0.20 [56]	8.3 ± 0.1 5.27 [57] 6.4 [56]	1.74 ± 0.05	0.88
Anion-Exchange Membranes				
AEM Type-I	0.14 ± 0.03	10.9 ± 0.1	1.60 ± 0.05	1.52
AEM Type-II	0.11 ± 0.02	3.9 ± 0.1	1.14 ± 0.05	0.76
AEM Type-X	0.06 ± 0.02	8.3 ± 0.1	1.63 ± 0.05	1.05
AMX	0.09 ± 0.03 0.11 [46]	3.7 ± 0.1 3.72 [46] 4.3 [77]	1.50 ± 0.05	0.62
MA-41	0.21 ± 0.03 0.24 [46] 0.18 [57]	4.2 ± 0.1 4.0 [46] 5.8 [57] 6.3 [56]	1.30 ± 0.05	0.58

**Table 5 membranes-09-00084-t005:** The values of real ($t_{1}^{*}$) and apparent ($t_{1app}^{*}$) counterion transport numbers, as well as the water transport number (*t_w_*) in the membranes under study at solution concentration of 0.5 mol dm^−^^3^.

Membranes	$t_{1}^{*}$	$t_{1app}^{*}$	*t_w_*
Cation-exchange Membranes			
CEM Type-I	0.985	0.950 [12]	3.9
CEM Type-II	0.996	0.976 [12]	2.2
CEM Type-X	0.996	0.994 [12]	0.2
CMX	0.995	0.98 [81] 0.965 [2] (NaCl)	1.7
MK-40	0.992	0.94 [36] 0.95 [82] 0.965 [83] (KCl)	5.8 6.5 [56]
Anion-Exchange Membranes			
AEM Type-I	0.989	0.950 [12]	4.3
AEM Type-II	0.995	0.970 [12]	2.8
AEM Type-X	0.997	0.970 [12]	3
AMX	0.996	0.98 [84]	1.9
МА-41	0.986	0.950 [85]	5.1 5.0 [56]

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
