# Peer review of "Transport Characteristics of Fujifilm Ion-Exchange Membranes as Compared to Homogeneous Membranes АМХ and СМХ and to Heterogeneous Membranes MK-40 and MA-41"

_membranes, 2019, doi:10.3390/membranes9070084_

Round 1
Reviewer 1 Report
This manuscript reports the structural and transport characteristics of EF-IEMs produced by Fjifilm, comparing with that of homogeneous and heterogeneous membranes. The results, in which electric conductivity, diffusion permeability and transport numbers were measured in this study, are useful for designing systems using ion exchange membranes. Furthermore, it is interesting that SEM and AFM images of the EF-IEM were used for explanation of transport characteristics. However, I have some minor comments. The publication of this manuscript has to be determined after clarification of remarks I attach below.
1. line 166: The sentence mentions about metallizing for obtaining AFM images of EF-IEM surfaces. I recommend to show the more detail method for metallizing in order not to confuse the reader.
2. Fig. 3 and 4 are not sufficient. If it is possible, please show the images of all EF-IEMs in Fig. 3 and 4. All additional images would help the reader understanding the structural and transport characteristics of EF-IEMs.
3. The images should be adopted the most common ones characterizing the membrane. Author needs to mention about reproducibility and dispersion for the images, especially in Fig 2 and 3.
4. line 245: How deep are holes mentioned at the sentence? Furthermore, did you observe these small holes on the surface of the other EF-IEMs? What roles do these holes play on transport characteristics?
5. line 246: Figure 4 shows the images of TypeⅠ. Does TypeⅡalso have macro pores?
6. line 270: This sentence interferes with the understanding of the first half of the next paragraph. I recommend that you reassess the composition of the sentences in Chapter 3.1.
7. The horizontal axis in Figure 6 is difficult to understand. How about replacing the top and bottom axes?
Reviewer 2 Report
The manuscript “Transport Characteristics of Fujifilm Ion-Exchange Membranes as Compared to Homogeneous Membranes АМХ and СМХ and to Heterogeneous Membranes MK-40 and MA-40” reported A comparative analysis of the concentration dependences of the conductivity, diffusion permeability, real and apparent ion transport numbers in cation- and anion-exchange membranes produced by three manufacturers. The authors have provided solid data to back up the conclusions in most cases. However, when reading the manuscript some questions arise, therefore some complementary information and revision should be taken into account before being published in “Membranes”.
1. The reviewer praises the efforts the authors have applied for this research. However, the Abstract and Conclusion parts in the manuscript appear to be too prolix and miscellaneous, which should comprehensively summarize the key points. Concise writing and appropriate deletion of some duplicate contents are beneficial for the readers to catch the essence of manuscript.
2. All the tables are not clear enough to show the data with one-to-one correspondence. Grid form tables are recommended.
Reviewer 3 Report
This manuscript reports the comprehensive characterization of some ion exchange membranes. The results contain some valuable results for researchers, but the manuscript is not well organized to read. The manuscript is too long with less-important and repetitive information. I point out some problems and comments as follows.
1. Line 106: The polymer shown in Figure 1 is “polyamide”, not “polyacrylamide.”
2. For readers, the authors had better summarize the structures and materials of membranes in Table.
3. Line 227: According to the caption of Figure 4, the images in Figure 4 are AEM Type I, but these are introduced as “CEM Type II” in the main text. Which is correct?
4. The authors discuss the effect of the protrusion of the fibers and the existence of the open-pore on the membrane property. Can you discuss these features using any physical value like “surface roughness”?
5. In Figure 5, the readers don’t need the images of (b) and (c). Instead of these images, the images of AMX and MA-41 membranes must be submitted.
6. SEM, AFM, optical images of all samples including AMX, CMX, MA-41, MK-40 membranes must be submitted as supplementary materials.
7. The authors must submit the standard contact porosimetry data of Fujifilm Ion-exchange membranes. These data would be important to confirm the existence of large pore in CEM and AEM Type I membranes.
8. Line 304: I guess “polysulfone” is wrong. “polyolefin” is correct.
9. Line 305: “polyacrylamide” must be changed to “polyamide.”
10. Line 314: Figure 7, not Figure 6.
11. The quantitative analysis of the secondary and tertiary amines using Cu2+ ion is not reliable method, because the coordination bond between the single amine group and Cu2+ is not so strong. AEM Type I membrane has many defect, so the membrane absorb Cu2+ ion deep inside, which cannot be removed by washing. According to Reference 31, the analysis based on the chemical shift of XPS N1s peak must be done.
12. The raw data of XRF analysis must be submitted as supplementary materials.
13. The symbol of the counter ion diffusion coefficient must be unified. D_i or D_+?
This manuscript will be accepted after major revision.
////////////////////////////////////////////////////////////
Round 2
Reviewer 3 Report
Although the authors did not conduct some additional experiments proposed in the previous reviewer report, I approve to accept this manuscript for publication.